# Cognitive Behavioral Therapy for Chronic Insomnia in Outpatients with Major Depression—A Randomised Controlled Trial

**DOI:** 10.3390/jcm11195845

**Published:** 2022-10-01

**Authors:** Henny Dyrberg, Bjørn Bjorvatn, Erik Roj Larsen

**Affiliations:** 1Department for Affective Disorders, Aarhus University Hospital, Central Denmark Region, 8000 Aarhus, Denmark; 2Department of Global Public Health and Primary Care, University of Bergen, Norway and Norwegian Competence Centre for Sleep Disorders, Haukeland University Hospital, 5009 Bergen, Norway; 3Mental Health Department Odense, University Clinic, Mental Health Service, J. B. Winsløws Vej 18, 5000 Odense, Denmark; 4Institute of Regional Health Research, University of Southern Denmark, 6700 Esbjerg, Denmark

**Keywords:** sleep disorder, sleep medicine, mood disorder, nonpharmacological treatment, insomnia severity index, dysfunctional beliefs and attitudes about sleep

## Abstract

The aim of this randomised controlled assessor-blinded trial was to examine the effect of cognitive behavioural therapy for insomnia on sleep variables and depressive symptomatology in outpatients with comorbid insomnia and moderate to severe depression. Forty-seven participants were randomized to receive one weekly session in 6 weeks of cognitive behavioural therapy for insomnia or treatment as usual. The intervention was a hybrid between individual and group treatment. Sleep scheduling could be especially challenging in a group format as patients with depression may need more support to adhere to the treatment recommendations. The primary outcome measure was the Insomnia Severity Index. Secondary measures were sleep diary data, the Dysfunctional Beliefs and Attitudes about Sleep Questionnaire, the Hamilton Depression Rating Scale, and the World Health Organization Questionnaire for Quality of Life and polysomnography. Compared to treatment as usual, cognitive behavioural therapy significantly reduced the insomnia severity index (mean ISI 20.6 to 12.1, *p* = 0.001) and wake after sleep onset (mean 54.7 min to 19.0 min, *p* = 0.003) and increased sleep efficiency (mean SE 71.6 to 83.4, *p* = 0.006). Total sleep time and sleep onset latency were not significantly changed. The results were supported by analyses of the other rating scales and symptom dimensions. In conclusion, cognitive behavioural therapy for insomnia as add-on to treatment as usual was effective for treating insomnia and depressive symptoms in a small sample of outpatients with insomnia and major depression. ClinicalTrials.gov Identifier: NCT02678702.

## 1. Introduction

When polysomnography is conducted, depressed patients seem to have problems with disrupted sleep continuity, reduced sleep depth and shortened REM sleep latency and increased REM sleep [1]. The majority of patients with major depressive disorders (MDD) report sleep quality complaints [2]. The relationship between insomnia and depression has been described as bidirectional [3]. Insomnia may double the risk of developing depression [4] and has been associated with a more severe presentation [5], suicidal ideation [6] and poorer treatment response [7]. Insomnia symptoms may persist after remission of depression [2] and seem to be a risk factor for depression relapse [8], although this has not been documented consistently [9].

Antidepressants offered for depression may affect sleep in different ways. The exact mechanisms of antidepressants on sleep are still not fully understood. It seems that some medications reduce REM sleep (Nortriptyline, SSRI (Selective Serotonin Reuptake Inhibitor)) or increase slow wave sleep (Amitriptyline, Mirtazapine) [1]. Adjunctive pharmacological treatment for insomnia in patients with depression often includes z-hypnotics, sedating antidepressants (e.g., mirtazapine, amitriptyline) or sedating antipsychotics [10]. However, these medications may produce daytime sedation and result in daytime napping, preoccupation about fatigue and difficulties falling asleep at night [11]. Patients with depression are often given sleep hygiene advice, but this has proven ineffective as a single intervention, especially among patients with chronic insomnia [12]. Standard cognitive behavioural therapy for depression does not address insomnia specifically. Standard cognitive behavioural therapy may be beneficial to other conditions such as children suffering from migraine [13].

Research has supported the efficacy of cognitive behavioural therapy for insomnia (CBT-I) without comorbidity [14]. According to a meta-analysis [15], CBT-I with comorbid medical or psychiatric disorders had medium to large effect on subjective sleep variables.

Several studies have investigated whether CBT-I may reduce depressive symptoms. One meta-analysis found that CBT-I seems to be effective and safe for insomnia comorbid with depression to improve the insomnia condition, while it is unsure whether CBT-I could improve the depression condition [16]. Two other meta-analyses included all CBT-I studies to determine whether treatment of insomnia leads to improved depression outcomes in individuals with both insomnia and depression and found an improved depression outcome with a medium to large effect size [17,18].

Support for the effect of individual face-to-face CBT-I on depressive symptoms has been found [19,20,21], but the effect of group treatment on depressive symptoms was unclear. It is uncertain to what extent patients with a clinical diagnosis of depression would benefit from CBT-I group therapy. Patients with depression may need more support than is obtainable in a regular CBT-I group setting. Especially sleep scheduling could be a challenge in a group format as patients with depression may need more support to adhere to the treatment recommendations [11].

The aim of our study was to investigate the effect of CBT-I delivered as a hybrid between individual and group format on insomnia and depressive symptomatology in outpatients with depression compared with treatment as usual (TAU). Our main hypothesis was that CBT-I would produce a greater improvement than TAU on insomnia severity and sleep variables measured by sleep diary and polysomnography. Furthermore, we hypothesized a more pronounced reduction in depression severity in the CBT-I group than in the group receiving TAU.

## 2. Methods

### 2.1. Study Design

The present prospective, randomised controlled trial study of cognitive behavioural therapy for insomnia (CBT-I) was offered as a multi-component intervention that included (i) sleep restriction; limiting time in bed to consolidate sleep and increase homeostatic sleep drive, (ii) stimulus control to promote a strong connection between sleep and bed, and (iii) cognitive therapy to address dysfunctional beliefs about sleep. Relaxation techniques were added to reduce physical arousal, and sleep hygiene education was included to improve sleep habits. We used a parallel design with 1:1 allocation ratio between intervention group and TAU group. We performed and described the trial in accordance with the CONSORT (Consolidated Standards of Reporting Trials) 2010 statement [22].

### 2.2. Eligibility Criteria

The 47 participants included were recruited between August 2015 and July 2018 in the Central Region, Denmark, from an outpatient clinic for patients with depression; from general practitioners, psychiatrists with private practice, social workers; and via newspaper advertisements. At baseline, participants who provided informed consent underwent a semi-structured face-to-face assessment interview with a clinical psychologist which included collection of demographic data (Table 1), screening for sleep disorders, the Mini-International Neuropsychiatric Interview (M.I.N.I. 5.0) [23] and a rating of depression severity by the 17-item Hamilton Rating Scale for Depression (Ham-D17) [24].

The M.I.N.I. (5.0) was administered to assess major depressive disorders (single or recurrent episode) and to exclude patients with substance abuse, high risk of suicide, bipolar disorder, or schizophrenia.

All participants underwent an initial one-night ambulatory polysomnographic sleep recording (PSG) to rule out sleep apnea and periodic limb movements (PLMS-index >15) and to obtain baseline data. This was performed 1-2 weeks before treatment commenced.

### 2.3. Inclusion Criteria

Aged 18–67;Major depression, single or recurrent episode [25] with a score above 17 on the 17-item Hamilton Depression Rating Scale;Sleep Onset Latency (SOL) or Wake After Sleep Onset (WASO) lasting more than 30 min or early morning awakenings at least three nights a week despite sufficient opportunity to sleep and impaired daytime functioning;Insomnia for at least three months, but some participants had suffered from insomnia for years [26].

### 2.4. Exclusion Criteria

Medical disorders considerably affecting sleep;Schizophrenia or bipolar disorders;Ongoing psychological treatment;Suicidality equivalent to level three on the 17-item Hamilton Depression Rating Scale;Current substance abuse;Pregnancy;Working at night shifts;Unable to speak or understand Danish;Other sleep disorders (e.g., severe sleep apnea (AHI (apnea-hypnopnea index) above 14) or restless legs syndrome) were excluded.

### 2.5. Randomisation and Blinding

Study data were collected and managed using Redcap electronic data capture tools. Participants were randomly assigned to either CBT-I plus TAU or TAU alone using computer-generated randomized blocks within age and gender strata.

Assessors of depression severity and sleep technicians were blinded to the participants’ treatment conditions. Patients were instructed not to disclose their intervention group.

Assessors were experienced nurses, technicians, psychologists, and students of psychology in their final study year. A flow chart describing the period from screening to completion of participants’ treatment is presented in Figure 1. Participants randomized to TAU alone were offered CBT-I treatment after the study had finished, but these data are not included in the present analyses. No payment was offered for study participation. No changes to the methods after trial commencement have been made.

### 2.6. Interventions

#### 2.6.1. TAU

In the TAU-alone condition, participants met individually with the general practitioner every third week or with the nurse at the same outpatient clinic to discuss symptoms of depression and to evaluate the effect of pharmacotherapy. Besides monitoring of suicidality guidance to cope with problems in daily life was offered. Participants were treated according to Danish guidelines and medical recommendations for pharmacological treatment of unipolar depression [27] allowing for clinical judgement and flexibility for first choice and switching of antidepressant medication. Participants were not allowed to use modafinil or the like due to its stimulating effect. TAU sessions could include discussion of sleep hygiene principles, but no stimulus control or sleep restriction was applied. Participants were allowed to change their antidepressants if their general practitioner or psychiatrist (with private practice or in the outpatient clinic at the hospital) recommended it.

#### 2.6.2. CBT-I

The experimental group received CBT-I in addition to TAU. Participants requested to participate in the CBT-I group were allocated to one weekly treatment session for six weeks. Treatment took place in a psychiatric outpatient clinic at the University Hospital. The intervention was a hybrid between individual and group treatment. Especially sleep scheduling could be a challenge in a group format [11]. Therefore, a hybrid between group and individual treatment was chosen for the intervention. The first session included behavioural therapy components and an individual sleep schedule was established [11]. The first individual 60 min session was followed by five group sessions lasting 90 min each. The group consisted of 3-5 participants. The treatment duration was six weeks. Group membership was fluid, with participants leaving when they had attended all group sessions, and new members continuously joining the group. By this we wanted to avoid a waiting list to the groups to reduce the risk that participants might lose motivation to participate. The CBT-I intervention was conducted by the first author, an experienced, licensed cognitive therapist trained through CBT-I seminars with Michael Perlis, Don Posner, and Jason Ellis. Seminars included supervision. The treatment manual was based on material from existing CBT-I manuals [28,29]. The treatment manual included a checklist for each session filled out by providers to ensure that all issues were covered.

Each group session followed a standard schedule: welcoming new members; 15 min of progressive muscle relaxation training [30] conducted by a physiotherapist with more than 10 years of experience with treatment of mood disorders; setting of individual goals for new group members and evaluating the goals and homework assignments of other members including troubleshooting of problems with adherence; working with the theme of the session using education, sharing and dialogue; and finally, giving a new homework assignment. Co-therapists were psychiatric nurses with more than 10 years of experience with treatment of mood disorders. Participants stayed for ten minutes after the group session together with therapist or co-therapist to individually calculate sleep efficiency and agree to a new prescribed bedtime following the principles of sleep restriction [28]. Information handouts and homework worksheets written in lay terms were provided along with a workbook. A written summary of the group session made by the co-therapist was sent to participants to compensate for cognitive problems (attention and working memory) often experienced by patients with depression and insomnia. Participants were allowed to change their antidepressants if their general practitioner or psychiatrist recommended it.

#### 2.6.3. Measures

Participants in the CBT-I and TAU group were given a two-week sleep diary, and the questionnaire Insomnia Severity Index, the Dysfunctional Beliefs and Attitudes about Sleep questionnaire [29] and WHO-5 Subjective Quality of Life Scale [31]. All participants completed the same outcome measures at baseline and after six sessions of CBT-I treatment plus TAU or six weeks of TAU. We only used validated rating scales and did not use any further questionnaires.

A recording of PSG to assess post-treatment sleep change was performed in the two weeks after the end of treatment. The equipment was set up in a sleep lab according to the guidelines of the American Academy of Sleep Medicine (AASM) manual [32] late in the afternoon, and participants went home to sleep in their own beds. The PSG was recorded using a 24 channel XLtec Trex HD ambulatory headbox (Natus Medical Incorporated, 2630 Taastrup, Denmark) including six channels of electroencephalography, electro-oculography, electrocardiography, surface electromyography of the submental and tibial muscles, nasal airflow, respiratory inductance plethysmography and pulse oximetry. The recordings were analysed with the SleepWorks software (Natus Medical Incorporated, https://natus.com/products-services/natus-sleepworks-software (accessed on 28 August 2022)). An experienced technician trained in sleep scoring visually scored the sleep recordings according to the AASM criteria version 2.2 (2015), and the scoring was checked by a neurologist specializing in sleep medicine. Both were blinded to the allocation of the participants. Main PSG outcome measures included total sleep time (TST), time in bed (TIB), sleep onset latency (SOL), wake after sleep onset (WASO), sleep efficiency (SE = TST/TIB (× 100), and arousal index.

Participants completed a sleep diary including items from the sleep consensus diary [33] during a two-week period before and after treatment. For each 24 h period, participants noted their estimates of bedtime, sleep latency, number and duration of awakenings, early morning awakenings and rise time. Sleep efficiency TST/TIB (× 100) was calc ulated. Naps, use of medicine, alcohol, caffeine, and nicotine were registered.

The Insomnia Severity Index (ISI) [29] was translated into Danish by permission of the author. The seven-item self-report questionnaire measures the participant’s subjective symptoms and consequences of insomnia. The total score ranges from 0 to 28 with higher scores representing more severe insomnia symptoms. A score in the 0–7 range indicates no clinically significant insomnia. Scores in the 8-14 range indicate sub-threshold insomnia and scores in the15–28 range indicate moderate to severe levels of insomnia. The ISI has been evaluated with adequate psychometric properties and is used to measure changes in perceived sleep difficulties. A change score of −8.4 was associated with moderate improvement in a clinical sample [34]. Even so, a six-point reduction in ISI was recommended as a clinically meaningful improvement in the treatment of primary insomnia [35]. The Cronbach’s alpha in our study was 0.54.

The HAM-D17, 17-item version [24], is a validated depression measure developed for follow-up treatment for depression. Patients are asked about their mental, emotional, and physical state during the past three days. The total score ranges from 0 to 52. To assess core depressive symptoms, the score on the Hamilton 6-item version was also calculated [31].

The Dysfunctional Beliefs and Attitudes about Sleep (DBAS-16) questionnaire is designed as a self-report measure with 16 items addressing common negative thoughts about sleep. It measures participants’ concerns about insomnia in general on four subscales: consequences (item 5,7,9,12,16) worry/helplessness (item 3,4,8,10,11,14), expectations (item 1,2), and proneness towards sleep medication (item 6,13,15). “Consequences” include exaggeration of disturbances in functioning due to the sleep problems. “Worry/helplessness” include exaggeration of losing control and impact on health. “Expectations” include a belief that a certain amount of sleep is necessary. “Medication” includes the opinion that medication must solve the problem. Dysfunctional beliefs and attitudes may potentially contribute to cognitive arousal perpetuating insomnia. The DBAS-16 is formatted as a ten-point Likert scale; 0 indicates strongly disagree and 10 indicates strongly agree. Higher scores indicate more endorsement of dysfunctional beliefs and attitudes about sleep. Adding scores for all 16 items and dividing by 16 yields an average total score. Likewise, the mean of subscale scores can be calculated. The DBAS-16 has been found to be sensitive to changes after insomnia treatment. It has proven to have stability over time and possesses a high internal consistency [36]. In our study, the Cronbach’s alpha was 0.93.

Participants were asked to complete the WHO-5. A subjective quality of life questionnaire [31] to assess daytime functioning [37]. The Cronbach’s alpha in our study was 0.91.

### 2.7. Ethics and Registration

The project was approved by the Danish Central Region Committee on Health Research Ethics (no. No. 1-10-72-35-15) and by the Danish Data protection Agency (no. 1-16-02-413-15). ClinicalTrials.gov Identifier: NCT02678702. Eligible participants were introduced to the aim, procedures, perspectives and the possible discomforts or benefits of the study. They signed a standard informed consent form describing their rights to withdraw at any time during the study without this having any consequences for their treatment in general. Staff members have received CGP training.

### 2.8. Statistics

A power calculation was completed before the study was initiated. In a validation study of persons referred to a sleep clinic [38], a mean ISI of 19.7 and a standard deviation (SD) of 4.1 was found. The calculation of SD was based on empirical data and thus holds a slight uncertainty. The minimal important difference (MID) was set as a reduction in ISI on four corresponding to 20%. We needed 20 patients in each group to achieve an 85% chance of finding a difference between the groups at a 0.05 significance level. If discontinuation was about 20%, we would need a minimum of 24 participants in each group, with a total of 48 participants.

The demographic data (sex, civil status, employment, etc.) were presented using descriptive statistics for all participants (Table 1) and stratified in the two treatment groups: TAU and CBT-I. For comparison of the demographic data between the two groups, *p*-values were based on Fisher’s exact test for categorical variables [39] and on Student’s two-sided *t*-test for continuous variables [40]. A normality check of data was performed using Shapiro–Wilk’s test [41] and a visual inspection of their histograms and normal Q-Q-plots as well as evaluation of the skewness and kurtosis’ z-values. We were unable to reject the null hypothesis that the data were normally distributed.

To compare the differences of treatment effect within the TAU and CBT-I group at baseline and after 6 weeks with two measurements we used Student’s two-sided paired sample *t*-test for continuous variables. To compare the differences of treatment effect between the TAU and CBT-I group at baseline and after 6 weeks with two measurements we used Student’s two-sided unpaired sample *t*-test for continuous variables. As the last estimation between groups are not unaffected by baseline differences, we chose to add analyses of covariance that generally has greater statistical power (ANCOVA) [42]. The method is a regression method and use a change score by subtracting the follow up change score from the baseline score. The model makes it possible to include additional prognostic variables, but the demographic Table 1 did not reveal a need for this. Concerning the daily measurement of SE in 14 days from the sleep diary before and after the intervention we performed mixed effects linear regression models (LME) [43] with full information maximum likelihood estimation. These models make it possible to include all relevant covariates across repeated measurements and efficiently handle missing data. We included subgroups concerning gender and age as well. We applied the Bonferroni correction. The datasets analysed during the current study are available from the corresponding author on reasonable request.

## 3. Results

### 3.1. Baseline Patient Characteristics

A total of 47 participants were randomised; 6 participants from the CBT-I group (*n* = 23) and 5 from the TAU group (*n* = 24) had received cognitive behavioural therapy for depression during the past year. In the total sample, 35.9% of the participants had a comorbid psychiatric diagnosis, mostly anxiety disorders, in addition to depression. Demographic and clinical characteristics of participants are shown in Table 1. No statistically significant differences in demography were recorded between the two groups at baseline.

### 3.2. Completion and Attendance

One participant in the CBT-I group withdrew before the intervention started (Figure 1). Participants in the CBT-I intervention group (*n* = 22) attended from two to six sessions. Six participants in the CBT-I group dropped out during treatment: mean weeks of participation for the six participants was 3 weeks. One stopped due to worsening of anxiety and five because they could not tolerate restrictions in their sleep schedule.

Participants dropping out from the CBT-I group are compared to completers (Table 2). They had a mean age of 25.17 versus 40.13 among completers (*p* = 0.001), a mean TST of 470.54 versus 378.45 (*p* = 0.007) but no significant differences in sex, Ham-D17, ISI, SE, SOL or WASO. Among participants in the TAU group, (*n* = 24) four persons regretted to participate in the follow-up because they could not find the time and one was hospitalised. They did not differ significantly from the completers in the TAU group.

### 3.3. Sleep Outcome Measures

Before the intervention at baseline, no significant differences in sleep parameters from the Sleep Diary were observed in the between group comparison of the CBT-I and the TAU group (Table 3).

After the intervention at follow up, the means within groups are compared.

Within the CBT-I group the means in sleep parameters improved significantly: SOL (58.9 to 36.1, *p* = 0.002), WASO (54.7 to 19.0, *p* = 0.002) and ISI (20.6 to 12.1, *p* = 0.001). In the TAU group, no significant change within the group was found.

In the between-group comparison, no significant difference in mean sleep parameters was found (TST, SOL, WASO). In the comparison of change, WASO was significantly reduced in the CBT-I group (*p* = 0.003), whereas SOL and TST were not.

After the intervention at follow up, the mean ISI between groups changed significantly (Table 3) and (Figure 2) when CBT-I are compared to TAU: ISI (12.1 vs. 17.7, *p* = 0.01). The change score in ISI was found to be significant as well (−8.1 vs. −3.1, *p* = 001).

Concerning the daily measurement of SE in 14 days from the sleep diary before and after the intervention age and gender were included as fixed effects in the mixed effects model and were not significant. ID was chosen as random effect and time, group, and interaction between group as fixed effects. We applied the Bonferroni correction. From the mixed effects linear regression models (LME) of repeated sleep efficiency measurements we found a significant treatment effect with increased mean sleep efficiency of CBT-I (from 71.6% to 83.4%) vs. TAU (69.7% to 72.1%); (*n* = 41, β = 1.32, Std. error 2.82, df 40.502, *t* = 467, *p* = 0.50, 95% CI −4.38 to β = 10.66, Std. error 3.09, df 35.470, t = 3.449, 95% CI 4.39–16.93, *p* = 0.006).

#### Data from PSG Recordings

Before the intervention, no significant differences in sleep parameters were observed between the CBT-I and the TAU group. After the intervention the within group analysis showed no significant changes in the CBT-I group. In the TAU group, SOL was significantly reduced in the polysomnography measurements (from 23.4 to 10.2, *p* = 0.02). The difference in change score of SOL between the groups was, however, not significant. The SOL started at a much higher baseline level in the TAU group compared to the CBT-group (23.4 vs. 15.5). Therefore, it may be important to include the change score that is not dependent on the baseline level.

In the between-group comparison, no significant change was found. In the ANCOVA model, the change score in WASO was significantly reduced in the CBT-I group (−37.0 vs. −4.1, *p* = 0.003).

### 3.4. HAM-D17, HAM-D6 and WHO-5

Before the intervention, no significant differences were found between the groups (Table 4). After the intervention, the within-group analysis found a significant treatment effect in the CBT-I group of HAM-D17 (21.2 to 13.8, *p* = 0.001), HAM-D6 (9.5 to 7.4, *p* = 0.050) and WHO-5 (25.3 to 39.2, *p* = 0.01). In the TAU group, no significant difference in within-group means was found. The between-group comparison found a significant reduction in means in HAM-D17 (*p* = 0.04). In the ANCOVA model, the change score was significant in HAM-D17 (*p* = 0.003) and WHO-5 (*p* = 0.003), but not in HAM-D6 (*p* = 0.16).

### 3.5. DBAS-16

Results of DBAS-16 pre-and post-treatment for the two treatment conditions are shown in Table 5.

Before the intervention, no significant differences were found between the groups. After the intervention, the within-group analysis found a significant treatment effect in the CBT-I group for the total score of DBAS (101.4 to 72.8, *p* = 0.002) as well as for the subscales “consequences” (34.2 to 27.4, *p* = 0.04), “helplessness” (42.8 to 30.4, *p* = 0.003), and “expectations” (12.8 to 7,6, *p* = 0.005). In the TAU group the total DBAS score decreased as well (102.3 to 93.1, *p* = 0.036), as did “consequences” (36.0 to 32.6, *p* = 0.047). In the between group comparison of CBT-I versus TAU a significant treatment effect of CBT-I was found for total DBAS (72.8 vs. 93.1, *p* = 0.02) and “medication” (7.3 vs. 12.6, *p* = 0.04). In the ANCOVA model a significant change score was found in the CBT-I group versus the TAU group in total DBAS (*p* = 0.001), in “consequences” (*p* = 0.004), in “helplessness” (*p* = 0.001) and in “expectations” (*p* = 0.003).

## 4. Discussion

CBT-I for patients with chronic insomnia and major depression resulted in a significant improvement in the intervention group in sleep efficiency, the severity of insomnia (ISI), quality of life (WHO-5) and in the symptom score of depression (HAM-D17). Particularly, our findings showed that it is possible to reduce symptoms of chronic insomnia with an intervention added to treatment as usual.

The improved sleep efficiency may, to some extent, reflect that we used sleep restriction. This may explain why sleep duration (TST) was reduced in the intervention group still practicing sleep restriction at follow-up. WASO was significantly reduced following CBT-I in both our measurement from the sleep diary and from the polysomnography. Disrupted sleep continuity is a severe problem among patients with a depression. A reduction in WASO improves sleep continuity, and the patients may experience reduced worry about not being able to sleep properly. Participants’ subjective evaluation of sleep severity expressed as an ISI change score of −8.1 in the present study has been found to indicate a moderate improvement in clinical samples [34].

In the TAU group SOL was significantly changed in the polysomnography measurements (from 23.4 to 10.2, *p* = 0.02). The difference in change score of SOL between the groups was, however, not significant. The SOL started at a much higher baseline in the TAU group compared to the CBT-group (23.4 vs. 15.5). Therefore, it may be important to include the change score that is not dependent on the baseline level.

The significant reduction in HAM-D17 between groups but not in HAM-D6 may be due to the inclusion of sleep items in the HAM-D17. The significant increase in quality of life (WHO-5) may also reflect a reduction in depressive symptoms; besides that, the participants felt more rested. The reduction in depressive symptoms is in line with previous findings [17,18]. However, other studies [16,44] found no convincing impact of CBT-I on depression. Despite a small sample size, the present study showed some improvement for patients diagnosed with moderate to severe levels of depression.

The clinical implications of this study suggest augmenting treatment for depression with CBT-I. Treating sleep problems alongside depression may increase the likelihood of remission [9]. However, non-adherence related to characteristics of depression such as anhedonia, rumination and maladaptive beliefs may challenge concomitant treatment [3]. Concerning dropouts from the intervention group primarily among the younger participants give rise to reflections concerning the methods. Sleep restriction demands regularity, and this may not apply well to young patients. Perhaps the younger participants would prefer an internet-based intervention or a sleep diary delivered as an app [45]. Baseline sleep duration among the dropouts was 470 min compared to 378 min among the completers. Although the dropouts have registered a lower SE of 73.8%, the normal sleep duration may have influenced their motivation for treatment and their experience of the benefit. Reasons for drop-out in the control group were not related to treatment with CBT-I but may be due to disappointment of not being offered CBT-I.

The significant reduction in scores regarding dysfunctional beliefs and attitudes about sleep in the intervention group may reflect how the intervention has worked by improving coping with insomnia. Regarding the score of the subscale, “consequences” was reduced in the within TAU group comparison as well. However, the change was significantly higher in the CBT-I group. As the TAU group was instructed to complete a sleep diary for four weeks, they may be aware of that they are able to function without a good night’s sleep. This could explain the reduction in the score “consequences”. However, we did not see a reduction in the subscales worry and helplessness or expectations. Overall CBT-I changed participants’ appraisal of insomnia. Changing dysfunctional cognitions about sleep may lead to improved sleep or improved sleep may alter beliefs. The mechanisms of change are unknown but when levels of worry are reduced through successful CBT-I treatment it may also change the appraisal of insomnia severity.

Our study has several strengths. The study sample was a homogeneous group with clinically diagnosed moderate to severe depression and comorbid chronic insomnia, which contributed to a clear study focus. Another major strength was that we used both subjective (diaries, questionnaires) and objective (polysomnography) outcome measures to evaluate the effect of the intervention. Furthermore, we used validated and recommended measures such as ISI, DBAS and HAM-D17.

The study had several limitations. Due to problems recruiting patients, the sample size was smaller than the a priori power analysis recommended. The power analysis was based on ISI, which may lower the possibility to show an effect of CBT-I on sleep parameters such as SOL. Missing data could not be retrieved due to dropouts and inability to fill in the sleep diaries. The sampling of data might be more efficient if it were internet-based, where you may be able to give reminders to the participants.

A greater proportion of dropout from the CBT-I group might have led to a more motivated sample, but compared to the total CBT-I group, no significant difference in depression severity was found. Furthermore, contact with healthcare providers was less frequent in the TAU group than in the CBT-I group. We are unable to rule out the option that more frequent therapeutic contact influenced the results. Although the TAU group was offered an intervention of CBT-I after the project ended, it may have had an impact on the wish to participate in the follow up. Additionally, we did not perform a two-night PSG recording or included longer follow-up due to economic constraints. Thus, we do not know if the positive findings persisted over time. A limitation is that we recorded medication at baseline and no change during the intervention. These changes could have had an impact on sleep. Participants were not blinded to allocation, and this may have influenced the answers. These limitations need to be taken into consideration when interpreting the results.

## 5. Conclusions

CBT-I in patients with chronic insomnia and major depression resulted in a significant improvement in the intervention group for sleep efficiency, for the severity of insomnia (ISI), for quality of life (WHO-5), and for the symptom score of depression (HAM-D17).

## Figures and Tables

**Figure 1 jcm-11-05845-f001:**
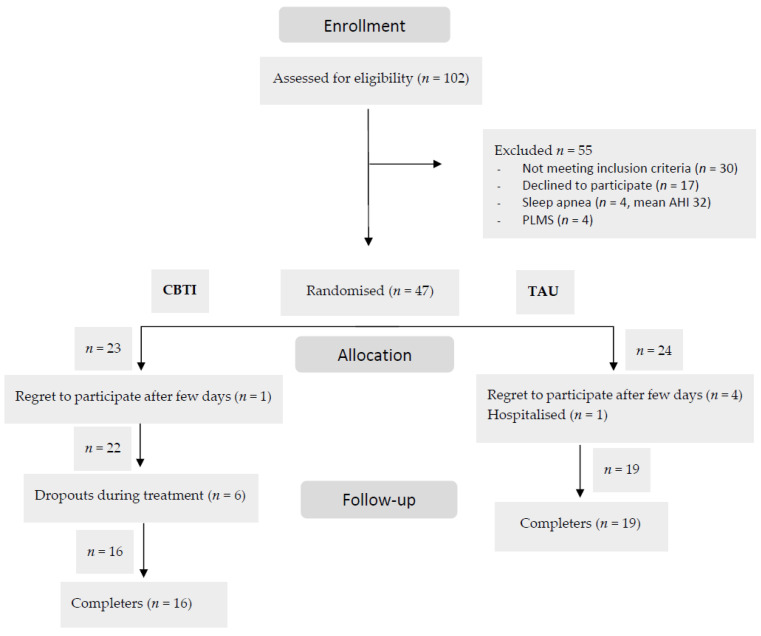
Participant Flow Diagram. AHI: Apnea-Hypopnea-Index. PLMS: Periodic Limb Movements. CBTI: Cognitive Behavioural Therapy of Insomnia. TAU: Treatment-as-usual.

**Figure 2 jcm-11-05845-f002:**
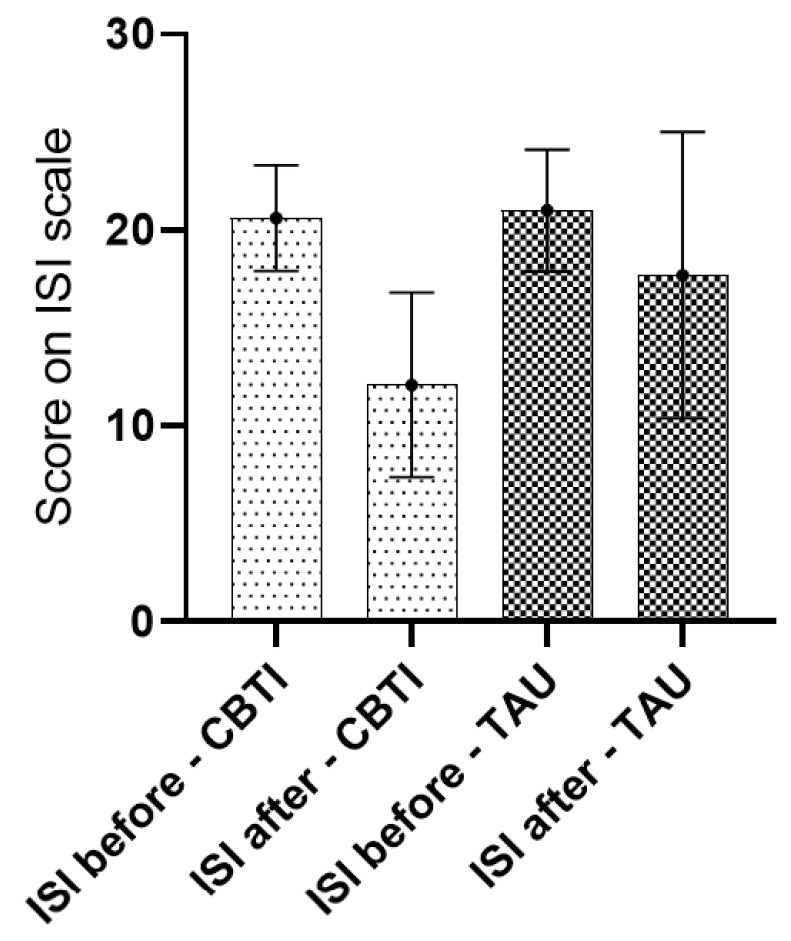
Insomnia severity index at baseline and after 6 weeks of treatment with CBTI or TAU.

**Table 1 jcm-11-05845-t001:** Demography of clinical study data.

Baseline Characteristics	All (*n* = 41)	TAU (*n* = 19)		CBT-I (*n* = 22)		*p*-Value
Sex Female	28 (68.3%)	14 (73.7%)		14 (63.6%)		0.52
Age Mean (Range)	37.07 (20-67)	38.26 (20-67)		36.05 (20-66)		0.62
Civil Status Living alone Living together	11 (26.8%) 30 (73.2%)	6 (31.6%) 13 (68.4%)		5 (22.7%) 17 (77.3%)		0.17
Employment Job Retired Student Unemployed	7 (17.7%) 6 (14.6%) 15 (36.6%) 13 (31.7%)	3 (15.8%) 3 (15.8%) 6 (31.6%) 7 (36.8%)		4 (18.2%) 3 (13.6%) 9 (40.9%) 6 (27.3%)		0.90
Years of Schooling 7–14 14–20	30 (73.2%) 11 (26.8%)	13 (68.4%) 6 (31.6%)		17 (56.7%) 5 (22.7%)		0.73
BMI (kg/m^2^) Mean (SD)	25.5 (6.7)	25.7 (8.3)		25.3 (5.1)		0.84
Height (cm) Mean (SD)	173.7 (10.0)	173.63 (11.1)		173.75 (9.2)		0.14
Weight (kg) Mean (SD)	76.73 (20.3)	77.4 (25.3)		76.14 (15.4)		0.84
Previous Depressive episode Yes	28 (68.3)	16 (84.2)		12 (54.5)		0.052
Use of Psychotropics Yes No	34 (83.0%) 7 (17.0%)	15 (78.9%) 4 (21.1%)		19 (86.3%) 3 (13.6%)		0.54
Medication Name of medication Dose range Some participants used more than one psychotropic		29 non-unique users		31 non-unique users		
		Sertraline	100–150 mg	Sertraline	100 mg	
		Escitalopram	20 mg	Escitalopram	30 mg	
		Citalopram	40 mg	Citalopram	40 mg	
		Venlafaxine	150–225 mg	Fluoxetine	60 mg	
		Duloxetine	60–120 mg	Venlafaxine	150 mg	
		Nortriptyline	100–325 mg	Duloxetine	30–120 mg	
		Clomipramine	1125 mg	Mirtazapine	30 mg	
		Mirtazapine	7.5–30 mg	Nortriptyline	100–150 mg	
		Quetiapine	25–50 mg	Valdoxane	25–50 mg	
		Melatonin	6 mg	Quetiapin	25–300 mg	
		Z-hypnotics	7.5–10 mg	Phenergan	25 mg	
		Chlorprothixen	50 mg	Melatonin	9 mg	
				Z-hypnotics	10 mg	
				Benzodiazepines	10–22.5 mg	

Student’s two-sided unpaired sample *t*-test is used for continuous variables and Fisher’s exact test for categorical variables. Non-unique users: participants may have been treated with more than one of the medications mentioned.

**Table 2 jcm-11-05845-t002:** Comparison of differences in means between dropouts and completers in the CBTI and TAU groups.

	Age (Years)	Sex Females %	HAM-D17	ISI	Sleep Efficiency Sleep Diary %	SOL Sleep Diary (min)	WASO Sleep Diary (min)	Total Sleep Time Sleep Diary (min)
Dropouts of CBTI, *n* = 6								
Mean	25.17	50%	20.33	21.80	73.81	84.32	27.68	470.54
SD	4.22		1.86	2.39	7.21	36.34	16.75	39.78
Completers of CBTI, *n* = 16								
Mean	40.13	63.6%	21.50	20.19	70.68	52.48	63.17	378.45
SD	14.62		6.02	2.80	10.53	29.42	51.50	63.65
Sign *	0.001	0.54	0.25	0.26	0.51	0.80	0.15	0.007
Df	19,511		20	19	20	18	19	19
Dropouts of TAU, *n* = 5								
Mean	29.0	80.0%	21.20	18.50	78.52	61.67	14.63	409.77
SD	9.8		3.03	3.54	7.80	34.05	12.95	66.78
Completers of TAU, *n* = 19								
Mean	38.3	73.7%	21.1	20.7	68.3	82.2	50.6	362.7
SD	13.9		2.5	3.0	9.4	57.8	36.8	128.4
Sign *	0.18	1.00	0.88	0.30	0.10	0.57	0.12	0.55
Df	22	1	19	19	16	17	17	16

* Student’s *t*-test is used for continuous variables between the dropouts and the completers. Fisher’s exact test is used for categorical variables.

**Table 3 jcm-11-05845-t003:** Means, standard deviations and significance levels among the TAU group and the CBTI group.

Measures	TAU	CBT-I	*p*-Value ^a^	*p*-Value ^a^	*p*-Value ^b^	*p*-Value ^c^
	Mean (SD)	Mean (SD)	within TAU Groups	within CBT-I Groups	between Groups	between Groups
	*n* = 19	*n* = 22	Baseline vs. Follow Up	Baseline vs. Follow Up	Baseline and Follow Up	Difference (Ancova)
Sleep diaries						
TST baseline	362.7 (128.4)	400.4 (70.5)			0.27	
TST follow up	361.5 (112.8)	379.3 (68.3)	0.88 (df14)	0.30 (df17)	0.57	
Change score	2.93 (70.4)	−21.14 (83.0)				0.50 (df 1)
SOL baseline	82.2 (57.8)	58.9 (32.6)			0.14	
SOL follow up	102.5 (131.9)	36.1 (38.1)	0.40 (df 15)	0.002 (df 17)	0.53	
Change score	29.2 (135.0)	−22.3 (25.2)				0.83 (df 1)
WASO baseline	50.6 (36.8)	54.7 (47.8)			0.78	
WASO follow up	43.8 (41.1)	19.0 (16.4)	0.58 (df 15)	0.002 (df 16)	0.27	
Change score	−4.1 (29.0)	−37.0 (41.5)				0.003 (df 1)
ISI baseline	21.0 (3.1)	20.6 (2.7)			0.64	
ISI follow up	17.7 (7.3)	12.1 (4.7)	0.08 (df 17)	0.001 (df 15)	0.01	
Change score	−3.1 (7.0)	−8.1 (5.6)				0.001 (df 1)
Sleep efficiency baseline ^d^	69.7 (SE 2.3)	71.6 (2.1)				0.50 (df 40705)
Sleep efficiency follow-up ^d^	72.1 (SE 2.8)	83.4 (2.8)				0.006 (df 34915)
Polysomnography						
TST baseline	417.9 (92.8)	421.0 (68.7)			0.91	
TST follow up	428.8 (99.7)	385.1 (52.5)	0.94(df 16)	0.09(df 15)	0.13	
Change score	1.8 (101.3)	−35.7 (78.4)				0.29 (df 1)
Sleep efficiency baseline ^d^	83.6 (8.5)	82.1 (10.9)			0.69	
Sleep efficiency follow up ^d^	82.0 (9.8)	85.8 (8.0)	0.61(df 16)	0.21(df 15)	0.24	
Change score	−0.99 (7.7)	4.0 (12.3)				0.40 (df 1)
SOL baseline	23.4 (20.4)	15.5 (12.1)			0.15	
SOL follow up	10.2 (12.7)	9.0 (8.1)	0.02 (df 16)	0.17 (df 15)	0.77	
Change score	29.2 (135.0)	−22.3 (25.2)				0.83 (df 1)
WASO baseline	57.7 (37.7)	70.0 (70.5)			0.5	
WASO follow up	81.1 (59.4)	49.3 (30.2)	0.15(df 16)	0.15(df 15)	0.06	
Change score	−4.1 (29.0)	−37.0 (41.5)				0.003 (df 1)
AHI baseline	2.2 (2.8)	3.3 (4.8)			0.38	
AHI follow up	2.5 (3.5)	2.8 (3.7)	0.99(df 14)	0.3(df 13)	0.8	
Change score	0.01 (3.9)	−1.3 (4.6)				0.42 (df 1)

^a^ Student’s two-sided paired sample *t*-test. ^b^ Student’s two-sided unpaired sample *t*-test. ^c^ Ancova model has been used to estimate and test the mean difference between groups. ^d^ 14 days repeated measures from the sleep diary before and after the intervention had missing data and was analysed with mixed effects linear models. SE: Std. error. SD: Std. deviation. TST: Total Sleep Time. SOL: Sleep Onset Latency. WASO: Wake After Sleep Onset. ISI: Insomnia Severity Index. AHI: Apnea-hypopnea index.

**Table 4 jcm-11-05845-t004:** Means, standard deviations and significance levels among the TAU group and the CBTI group.

Measures	TAU	CBT-I	*p*-Value ^a^	*p*-Value ^a^	*p*-Value ^b^	*p*-Value ^c^
	Mean (SD)	Mean (SD)	within TAU Groups	within CBT-I Groups	between Groups	between Groups
	*n* = 19	*n* = 22	Baseline vs. Follow Up	Baseline vs. Follow Up	Baseline and Follow Up	Difference (Ancova)
HAM-D6 baseline	9.8 (2.5)	9.5 (1.7)			0.61 (df 39)	
HAMD-6 follow up	9.8 (4.2)	7.4 (4.2)	0.71 (df 17)	0.05 (df 15)	0.10 (df 32)	
Change score	0.3 (3.7)	−2.3 (4.3)				0.16 (df 1)
HAM-D17 baseline	21.4 (2.9)	21.2 (2.1)			0.76 (df 39)	
HAM-D17 follow up	19.9 (9.4)	13.8 (7.0)	0.58 (df 17)	0.001(df 15)	0.04 (df 32)	
Change score	−1.2 (1.9)	−7.7 (2.0)				0.003 (df 1)
WHO-5 baseline	21.8 (10.2)	25.3 (11.6)			0.21 (df 37)	
WHO-5 follow up	30.3 (23.1)	39.2 (22.4)	0.1 (df 17)	0.013 (df 15)	0.27 (df 32)	
Change score	9.1	−22.3	15.1 (21.5)			0.003 (df 1)

^a^ Student’s two-sided paired sample *t*-test. ^b^ Student’s two-sided unpaired sample *t*-test. ^c^ Ancova model in SPSS has been used to estimate and test the mean difference between groups.

**Table 5 jcm-11-05845-t005:** Dysfunctional Beliefs and Attitudes about Sleep (DBAS-16). Means, standard deviations (SD) and significance levels in the TAU group and the CBTI group.

Measures	TAU	CBT-I	*p*-Value ^a^	*p*-Value ^a^	*p*-Value ^b^	*p*-Value ^c^
	Mean (SD)	Mean (SD)	within TAU Groups	within CBT-I Groups	between Groups	between Groups
	*n* = 19	*n* = 22	Baseline vs. Follow Up	Baseline vs. Follow Up	Baseline and Follow Up	Difference (Ancova)
Consequences						
Baseline	36.0 (9.5)	34.2 (8.4)			0.55	
Follow up	32.6 (10.1)	27.4 (10.2)	0.047 (df17)	0.040 (df14)	0.16	
Change score	−2.9 (5.8)	−5.6 (9.6)				0.004
Worry/Helplessness						
Baseline	40.2 (9.5)	42.8 (9.5)			0.37	
Follow up	36.3 10.0)	30.4 (9.3)	0.137 (df17)	0.003 (df14)	0.09	
Change score	−3.2 (8.8)	−10.7 (11.3)	0.001			
Expectations						
Baseline	12.7 (5.1)	12.8 (5.06)	0.94			
Follow up	11.6 (5.4)	7.6 (6.2)	0.362 (df17)	0.005 (df14)	0.06	
Change score	−0.78 (3.5)	−3.6 (4.2)				0.003
Medication						
Baseline	13.5 (7.3)	11.6 (9.2)			0.46	
Follow up	12.6 (7.3)	7.3 (6.9)	0.309 (df17)	0.232 (df14)	0.04	
Change score	−1.2 (4.9)	−1.9 (5.8)				0.108
Entire DBAS						
Baseline	102.3 (18.0)	101.4 (26.3)			0.9	
Follow up	93.1 (21.8)	72.8 (27.0)	0.036 (df17)	0.002 (df14)	0.02	
Change score	−8.2 (15.2)	−21.7 (21.9)				0.001

^a^ Student’s two-sided paired sample *t*-test. ^b^ Student’s two-sided unpaired sample *t*-test. ^c^ Ancova model has been used to estimate and test the mean difference between groups.

## Data Availability

The datasets analysed during the current study are available from the corresponding author on reasonable request.

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
