# Peer review of "Cognitive Behavioral Therapy for Chronic Insomnia in Outpatients with Major Depression—A Randomised Controlled Trial"

_jcm, 2022, doi:10.3390/jcm11195845_

Round 1

Reviewer 1 Report

Interesting report on CBT-I effectiveness on the whole.

Specific comments:

1. Please spell the abbreviation "SSRI" out in full in the first instance of its use.

2. As per established CONSORT guidelines, the RCT should be registered and the registry number presented; a full trial protocol should be available.

3. Please provide the actual institutional review board (IRB) approval/study number for the present study.

4. Some of the text in Figure 1 was truncated. Please adjust the alignment.

5. "Regret to participate after few days" - what exactly does this mean? Suggest elaborating on this.

6. Please change "a priory" to "a priori".

7. Please change "a real-life clinical setting" to "a real-world clinical setting". 

8. Please include a data availability statement.

Author Response

We very much appreciate all your comments and critiques. In the file added, we have listed your comments and our responses to these. Revised and added text in the manuscript is marked with red colour. 

Reviewer 2 Report

Dear authors, thank you for the opportunity to read your manuscript. The objective of the study is relevant and the methodology is appropriate to answer the research question. The article is generally well written and most of the aspects required are presented in the article. Some clarifications are needed about the missing data management and the blinding. You will find below recommendations that may improve the manuscript.

General suggestions

1. The use of "treatment as usual" is a bit confusing to me as there are many different ways to manage major depression. If antidepressants were the only treatment, I suggest changing "treatment as usual" to "antidepressant medications" in the whole article.

Title

I suggest that you clearly mention chronic insomnia in the title. The way it reads now is that you are trying to reduce insomnia symptoms in depressed patients. Randomised Controlled Trial should be capitalised like the rest of the title.

Abstract

1. You should clearly state the blinding of the trial in the abstract. The way I see it is that the trial was only assessor-blinded.

2. You should state clearly what was the primary outcome in the abstract.

3. The negative results should be stated in the abstract, not just the positive results.

4. Line 19, you use the abbreviation TAU but it is not used in the rest of the abstract. I suggest to remove "(TAU)"

5. Line 23 you use the abbreviation HAM-D17 but the abbreviation was not explained previously.

6. Line 25 It should be "effective" instead of "efficient".

7. The presentation of the results is a bit confusing. I suggest using one or two sentences to present all the results.

Introduction

1. I suggest removing the first two sentences of the introduction as they are not directly relevant to the topic (insomnia in people with depression).

2. Line 56-61 Please clarify if these studies targeted a people with depressive disorder or simply people with insomnia.

3. There is already significant evidence that CBT-I can help people with depression. You should present this evidence in more details.

4. It seems that the gap in the literature is the absence of hybrid format of CBT-I for people with insomnia and depression. I suggest to highlight this gap and link it with your objectives. If your goal was to test a hybrid format, this should also be stated in the abstract. You should also explain the potential advantages of this format over the formats that are known to be effective.

5. Please clarify which "support" is needed by people with depression (line 65)

Methods

1. Please use subsections such as "study design", "eligibility criteria" and "randomisation and blinding" to clarify the methods section.

2. Paragraph 3, you should specify who conducted the assessments and diagnosis during the screening.

3. Eligibility criteria. Please use number such as "1.", "2." to highlight the different criteria.

4. Clarify how "schizophrenia or bipolar" and "current substance abuse" were assessed.

5. Please clarify if the protocol was published, where and at what stage of the study.

6. Line 102, it should be technicians instead of technologists

7. Figure 1 should be clarified. I would not separate participants who dropped out before and after the start of the interventions. We should assume that they started the intervention at randomisation (when they where "prescribed" the intervention and accepted). Please clarify what you mean by "mean weeks of participation". If this is about the non-completers I suggest to remove it from the figure and place it in the text.

8. Line 141, in a clinical trial the participant are not "offered" treatment but "requested to participate".

9. Line 149. Do you mean that participation was conducted on a rolling basis with people at different stages of the treatment participating in the same session? If yes, please clarify the rationale.

10. Line 153 and 157, please clarify the term "experienced".

11. Line 149 to 154. You describe in detail the training by the first author but not the training of the nurses. Please clarify the roles in the intervention delivery of the therapist and co-therapists.

12. Line 166. Please clarify "cognitive problems", is these dysfunctional beliefs or reduced cognitive ability?

13. The TAU section should be combined with interventions. You should clearly state that both groups underwent TAU and the experimental group received CBT-I in addition to TAU.

14. You mention general practitioner or psychiatrist in line 168 and nurse or general practitioner in line 171. Please clarify.

15. Was the use of medication recorded during the trial? If yes, it should be reported in the methods and results sections. If no, this should be a limitation of the study.

16. You should report how many people underwent sleep hygiene in the control group.

17. Line 178-179 does "antidepressiva" mean "antidepressant"?

18. The Measures subsection is too long compared to the other subsections. Please remove unnecessary content.

19. Line 182 to Line 186. This should be in the eligibility assessment section.

20. Same for lines 192 to 197.

21. Line 206 technician, not technologist

22. Table 1 should be at the beginning of the results section

23. Please clarify how many people were using each anti-depressant drug.

24. Some participants used melatonin and z-hypnotics. I suggest conducting sensitivity analyses to explore the impact on the results.

25. You should clearly state if the questionnaires used in the study were validated. If yes, please add the reference. If no, please list it as a limitation.

26. Line 241 and 242. Not sure how practical is the blinding of assessors. I imagine that when the assessors asks about the symptoms of the patient, they may start to say that they improved since they took part in the CBT-I treatment, making the blinding ineffective. What measures were taken to ensure proper blinding? If none, this may be a limitation of the study.

27. Line 274. This subsection should be named Ethics and registration.

28. Please clarify in the ethics section if all the staff involved in the study received GCP training.

29. Line 287 and 288. Please provide a reference for the MID. The abbreviation does not match the full word.

30. Line 292, demographics were "presented", not "analysed".

31. The analysis is a bit complcated. You used t-test, ANCOVA and linear mixed model to assess efficacy outcomes. I suggest to use one statistics method only. Linear mixed model could be considered as it is better at managing missing data.

32. Lines 314 and 315. This belongs to the results section.

33. Line 329. This participant should be considered as a drop-out. 

34. Line 333. If the participant could not tolerate sleep restrictions, they should be proposed alternatives to sleep restriction. If this was not the case, this should be mentioned as a limitation.

35. Because of the large drop-out rate (around 30%) and the differences between the non-completers and completers, you should conduct missing data management to explore the impact of the missing data on the results.

36. The reasons for drop-out were different in the two groups. The drop-out for the experimental group are related with treatment and efficacy but the reason for drop-out in the control group seems unrelated. This should be discussed in the discussion section as it may affect the results.

37. At the begining of each paragraph you use one sentence about the baseline. This information should be placed at the begining of the results section, in the paragraph about baseline data.

38. Table 3. ISI is not part of the sleep diary.

39. It seems to me that many outcomes were not statistically significant. This should be stated more clearly in the results and discussion section, as well as in the abstract.

40. Figure 2 is redundant, we already have this information elsewhere.

41. The change in PSG-measured SOL in the control group should be discussed.

42. It seems to me that HAM-D6 was not mentioned previously.

43. Table 4 and table 5 should be combined with table 3. No need to report for each subsection of DBAS except if this is important to highlight a certain point. 

44. Did you record the changes in antidepressant medication during the study? This may also explain the changes in sleep experienced by participants. If not, this should be listed as a limitation.

Discussion

1. The discussion section should be reorganised. I propose the following paragraphs "summary of the results", "interpretation of the results", "comparison with existing literature", "strengths of the study", "limitations of the study" and "clinical implications".

2. Non-significant results should be discussed as well.

3. Please clarify "missing data could be me retrieved due to human error"

4. Please clarify "sampling of data may be more efficient if internet-based"

5. Another important limitation is the lack of blinding. You have used many participant-reported outcomes and the participants were not blinded to allocation. They may have reported better sleep outcomes simply because they thought this should be what is expected from them.

6. The real-life setting should not be considered as a limitation. The impact of changes in medication can be explored via statistical methods. If this was not the case, then this is a limitation.

Overall, the study is sound and the results are clinically useful. I hope the above suggestions will help the authors to improve the manuscript and that the article will benefit the field.

Author Response

(The authors gave the same response as above.)

Round 2

Reviewer 1 Report

Some outstanding comments and issues.

Specific comments:

1. In the introduction, authors should also mention that CBT has demonstrated generally good tolerability, acceptability and efficacy for other conditions e.g., migraines (citation: pubmed.ncbi.nlm.nih.gov/28028812).

2. The alignment for Figure 1 is off.

3. "... physiotherapist with more than 10 years of experience" - is this 'psychologist' and not 'physiotherapist'?

4. Table 1 should include information on the duration of sleep complaints. The group sampled in this study appeared to have particularly refractory or longstanding insomnia, given the relatively higher doses of melatonin they are receiving (usual doses being between 1 to 5mg).

Author Response

Response to Reviewer 1_second review

Thank you for reviewing our manuscript entitled: Cognitive Behavioural Therapy for Insomnia in Outpatients with Major Depression – a randomised controlled trial.

We very much appreciate all your comments and critiques. In the sections below, we have listed your comments and our responses to these. Revised and added text in the manuscript is marked with blue colour.

Specific comments:

  1. In the introduction, authors should also mention that CBT has demonstrated generally good tolerability, acceptability and efficacy for other conditions e.g., migraines (citation: pubmed.ncbi.nlm.nih.gov/28028812).

Our response: We have added a reference on this matter: Standard cognitive behavioral therapy may be beneficial to other conditions such as children suffering from migraine [13].  

  1. The alignment for Figure 1 is off.

Our response:

Thank you for this comment. We have discussed this with the editor’s office as we had problems with transfer of tables from Microsoft word to the JCM template.

  1. "... physiotherapist with more than 10 years of experience" - is this 'psychologist' and not 'physiotherapist'?

Our response: The sentence in the paper is correct. Both psychologists and a physiotherapist were involved in the study.

  1. Table 1 should include information on the duration of sleep complaints. The group sampled in this study appeared to have particularly refractory or longstanding insomnia, given the relatively higher doses of melatonin they are receiving (usual doses being between 1 to 5mg).

Our response: Thank you for this comment. It is difficult to achieve valid information on the exact duration of sleep complaints. We included participants with duration above 3 months and do not have estimations on a more exact duration.